# An Evaluation of the Wind Energy along the Romanian Black Sea Coast

Laura-Ionela Nedelcu [1,2,*], Viorel-Mihai Tanase [3] and Eugen Rusu [2]

1   Department of Navigation and Naval Transport, Faculty of Navigation and Naval Management, "Mircea cel Batran" Naval Academy, 1st Fulgerului Street, 900218 Constanta, Romania
2   Department of Mechanical Engineering, Faculty of Engineering, "Dunarea de Jos" University of Galati, 47th Domneasca Street, 800008 Galati, Romania
3   Maritime Hydrographic Directorate, 1st Fulgerului Street, 900218 Constanta, Romania
*   Correspondence: laura.nedelcu@anmb.ro

**Abstract:** The present study aims to outline a general overview of the wind energy potential along the Romanian coast of the Black Sea, using the weather data provided by the Maritime Hydrographic Directorate covering a 13-year time interval (2009–2021). The data obtained from seven automatic weather coastal stations distributed along the Romanian perimeter were used to evaluate the wind regime, highlighting the Black Sea's complex marine environment. The analysis based on the evaluation of the wind parameters per each station registered on the total period revealed that the overall wind characteristics are similar, resulting in no significant variations depending on the station's location. Moreover, the climatic picture of the Black Sea can be interpreted as two seasons, winter and summer, a conclusion based on the analysis made of the seasonal and monthly variation of the wind aspects. Subsequently, the outcomes obtained in this research imply that the Romanian Black Sea coast has the potential to be a good location for wind energy development due to the strong winds that blow in the region.

**Keywords:** wind potential; Romanian coast; Black Sea area; weather stations; marine environment

## 1. Introduction

Offshore wind energy is considered an important pillar in achieving net greenhouse gas emissions. According to the European Commission, by 2050, Europe must develop between 230 and 450 GW of offshore wind capacity, which will provide up to 30% of Europe's electricity needs. To date, Romania is the leader of Southeast Europe in terms of onshore development, with an installed capacity of 3 GW [1]. The onshore wind farm development was mainly carried out in the Dobrogea plateau, where almost 80% of the existing wind turbines in Romania are concentrated. The largest onshore wind farm in Europe is Fantanele–Cogealac, in the Dobrogea area, with an installed capacity of 600 MW [2].

Most research regarding the Black Sea wind and wave conditions indicates that the northwestern part presents a higher energy level than the eastern sector [3–8]. The evaluation carried out in [4], Rusu L., Bernardino M., and Guedes Soares C. outlined that the average wind speed value registered at Sulina meteorological station was 7.29 m/s, and at Chonomors'ke was 4.47 m/s, whereas at Gloria platform it was registered a value of 5.16 m/s. The observations provided by the two weather stations were made at a height of 10 m above the water, and those from the Gloria platform at a height of 36 m. In the same research, the wind speed analysis was made indicating that the western part registers significant seasonal variations, whereas the east is characterized by more stable conditions [9,10].

Furthermore, in their paper [10], the authors performed a comparison between the average wind speed from coastal stations and data provided by satellites. The analysis

stated the following wind speed average values: 4.9 m/s (coastal stations) and 6 m/s (satellites). Moreover, in [11], based on wind and wave conditions evaluated over a period of 50 years (1958–2007), it is concluded that the wind and wave climate of the western Black Sea basin presents a large spatial and seasonal variability. The authors mentioned that the wind speed exceeded 40 m/s offshore and 25 m/s near the shore.

The analysis carried out in [12,13] indicates that the Romanian coast has a higher energy level during the winter season, with an average wind speed of approximately 9.7 m/s and a power of 870 W/m$^2$ at a height of 80 m. Moreover, in [12] it is mentioned that during the warm season the wind climate registers values between 5.69–7.17 m/s with a maximum in September and a minimum in August. The cold season has an average wind speed located in the range of 7.67–8.59 m/s, with higher values being registered in January and minimums in October. The average annual wind speed at the height of 10 m is approximately 7.1 m/s.

Myslenkov et al. pointed out in the paper [14] that low-quality wind and wave potential were observed during the summer months. In their article [15], Rusu L., Răileanu A., and Onea F. obtained, based on the data provided by the Gloria platform, speeds of 6–7 m/s at heights of 10 m. In addition, in the papers [16,17], the authors highlighted the fact that in the vicinity of Romania and Ukraine, the wind speed during the winter season reaches an average value of 7.7 m/s and a maximum of 13.2 m/s. Diaconita A., Rusu L., and Andrei G. concluded in their study [18] that the average wind speed at a height of 10 m was 6.7 m/s. According to [19], the average wind speed is between 7–6 m/s in the offshore area and, respectively, near the Romanian coast. Near the Crimean Peninsula, an average wind speed of about 5.5 m/s was recorded.

Likewise, from the analysis carried out in the period 1979–2009, in [20], it was determined that in the December, January, and February season the winds have the highest magnitude, reaching values of approximately 8.1 m/s in the western part of the Black Sea and 8.7 m/s in the Sea of Azov. The authors specify that the Sea of Azov is exposed to strong winds compared to the Black Sea. The strongest wind values registered during summer (June, July, August) are in the Sea of Azov (6.1 m/s), whereas in the northwestern part of the Black Sea there are values around 5.7 m/s, whereas in the southwestern part, the wind reaches speeds of approximately 5.5 m/s. In spring (March–April–May), maximum wind speed values reach 7.0 m/s in the Sea of Azov and 6.3 m/s in the northwestern Black Sea. During autumn, maximum wind values of approximately 7.6 m/s were recorded in the Sea of Azov and 7.2 m/s in the northwestern Black Sea.

Based on the literature review mentioned above, the Black Sea area can be considered a valuable source of renewable energy. Data analyzed in [21] indicate that wind speed increases offshore, with only the central part of the deep-water sector having higher average wind speeds (nearly 7 m/s) compared to the southeastern part of the Exclusive Economic Zone (EEZ), where the wind speed decreases. A large part of Romania's EEZ has deep waters (>50 m), suitable for floating platforms. Whereas deep areas are accessible for floating wind turbines, at depths greater than 150 m the costs of maintenance increase significantly. Moreover, in [21], the authors evaluated the potential of the offshore wind sector in Romania, estimating a total capacity of 94 GW, which means a total annual energy production (AEP) of 239.04 TWh, of which 22 GW (AEP = 54.44 TWh) from fixed turbines and 72 GW (AEP = 184.6 TWh) from offshore wind farms.

This study can be considered an opportunity since there is a significantly growing interest in offshore wind project development on the Romanian coast of the Black Sea. Consequently, this research paper will focus on the analysis of the wind characteristics based on the data provided by the Maritime Hydrographic Directorate over a period of 13 years, indicating the wind potential along the Romanian coast. The elements of the novelty of the present work are defined as follows: (a) the Romanian coast wind regime is assessed by in situ measurements for seven sites located on or near the shoreline, compared to other research that used modeled or satellite data and very few observations; (b) a long-

term overview of the wind resources is provided (13 years); (c) the full wind characteristics coverage of the area because of the location of the stations.

## 2. Materials and Methods

### 2.1. Target Area

The Black Sea (Figure 1) represents an inland sea located between Europe, Anatolia, and the Caucasus, covering an area of 423,000 km². It has a volume of 555,000 km³, an average depth of 1315 m, and a maximum of 2258 m. It is considered one of the most isolated parts of the planetary ocean, being connected to the Mediterranean Sea through the Bosporus and Dardanelles straits and to the Sea of Azov through the Kerch Strait [22–24].

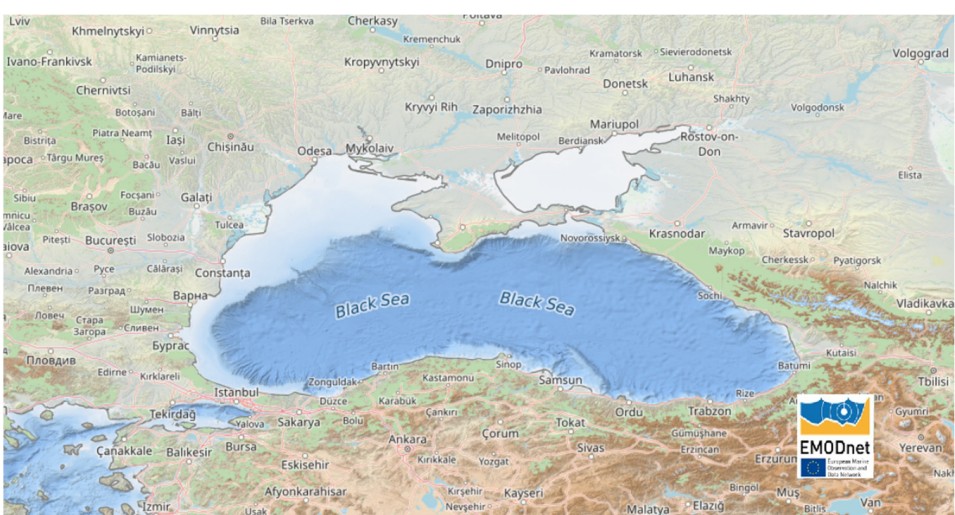

**Figure 1.** The Black Sea. Image obtained from [25].

The Black Sea is defined by an irregular shape, having as riparian states Romania, Ukraine, Russia, Georgia, Turkey, and Bulgaria. The largest bay is located in the western part of the sea off the coast of Bulgaria (bays of Bulgaria and Varna), followed by the bays of Odesa and Karkinit (in the northwest) and bays of Kalamit and Feodora in Crimea. In the eastern part of the sea can be seen the bays of Novorossiysk and Gelendzhik, whereas in the south there are Sinop bay and Samsun Bight. Crimea is the largest peninsula in the Black Sea [8].

The bathymetry of the Black Sea is divided between the reef, the continental slope, and the abyssal plain. The reef occupies up to 25% of the total area of the seabed and has average depths between 100–200 m and a width of 200 km. In the remaining area, depths below 2000 m can be found. The configuration of the bathymetric line as well as the presence of depressions and canyons have an important influence on the distribution of water masses and the direction and speed of currents [22–24,26].

The Black Sea region climate is temperate–continental, influenced by the baric centers situated above Southeast Europe: the Azores anticyclone and the Southeast European anticyclone. Furthermore, the Black Sea is roughly encompassed by mountain belts: the Pontics to the south, the Caucasus to the north and east, and the Carpathians to the west. Therefore, the sea climate can be indicated as follows: in the summer the air temperature is relatively uniform, whereas in the winter the air temperature can be described according to the geographical locations (minimums in the northwest corner and maximums in the southeast). Water temperatures vary according to the season: in summer, the surface water warms to 25 °C, (occasional extremes up to 28 °C), whereas in winter the sea temperatures reach 6–8 °C. The northwest coast has ice during winter, whereas the southeast remains around 9 °C [27,28].

One of the most valuable features of the Black Sea is that the water layers cannot mix. The sea's hydrochemical configuration is mainly influenced by basin topography and fluvial inputs, resulting in a strong stratified vertical structure. The first layer, located on the sea surface over 50 m, is less dense, less salty, and broadly cooler, fueled by large river systems. The second layer, the cold one, has a mass of water located at depths from 50 to 180 m. Fluvial inputs represent the European rivers that flow into the Black Sea (Danube, Dnieper, Don) as well as precipitation. Annually, the Black Sea gains an approximate quantity of 350 km$^2$ of water from the aforementioned rivers and nearly 250 km$^2$ from precipitation, whereas 350 km$^2$ of water is evaporated [22,26,29,30].

Black Sea salinity is established by the balance between freshwater received by rivers and the Mediterranean water entering through the Bosporus Strait. At the surface, an average salinity of 16–18 PPT (twice lower than the salinity of the surface waters of the world ocean) was recorded. The bottom layer salinity increases to 21–22.5 PPT at a depth below 120–200 m [31].

Two types of sea currents are present in the Black Sea: the surface currents, caused by the cyclonic wind pattern, and the double currents caused by the Kerch and Bosporus straits. Surface currents form two closed circles: the western and the eastern circles. The first one is opposite the Danube Delta and has a width of 100 km which decreases towards the south, whereas its velocity is around 0.5 km/h. The width of the second circle varies between 50 and 100 km, and the velocity is 1 km/h. The Bosporus double current consists of the water exchange between the Black Sea and the Marmara Sea. At the surface, there is less salty and lighter water that flows into the Marmara Sea at a speed of 2 m/s. At depths of 50–120 m, there is the saltier and denser water of the Marmara Sea which flows into the Black Sea with a speed of 4–6 m/s. Regarding the Kerch Strait, the first current, located at the surface flows from the Sea of Azov to the Black Sea with 1–2 m/s, whereas the bottom current flows in the opposite direction at a depth of about 5 m [22].

The Black Sea circulation is generally driven by the Rim current (cyclonic). This current follows the continental slope in the northwestern and western parts of the Black Sea and presents various cyclonic and anticyclonic whirlwinds within the primary current. The Black Sea has a horizontal circulation defined by gyres (cyclonic and anticyclonic currents directed by the force of the wind). Occasionally, the Rim Current has a width of tens of km and a maximum speed of 0.8–1 Nd or 40–50 cm/s, which sometimes reaches values up to 1.6 Nd or 80–100 cm/s [29,32–36].

The Black Sea tides vary from 1.1 cm near the Crimean Peninsula to 19 cm in the Dnieper–Bug estuary (Kherson and Nikolaev), 13–14 cm in Odesa and Illichivsk, and up to 12.6 cm on the eastern coast (Batumi). In general, semidiurnal tides prevail in the Black Sea [37].

*2.2. Wind Dataset*

The analysis carried out in this research aims at a comprehensive picture of the wind potential of the Black Sea northwestern part, the Romanian littoral, based on the data and information provided by the Maritime Hydrographic Directorate over a 13-year time period.

The Maritime Hydrographic Directorate is the Romanian Naval Forces specialized structure that provides meteorological, oceanographical, and climate information to the Navy's ships and units [38].

The department responsible for the data that this research relies on is the Climatology Laboratory from the Maritime Hydrographic Directorate (MHD). The purpose of this laboratory is to collect, manage, and analyze Meteorological and Oceanographic (METOC) data from the area of interest. Currently, the Naval Forces Marine Meteorological Surveillance Network (RSMM-FN), includes seven coastal automatic weather stations at the lighthouses subordinated to MHD and five naval automatic weather stations on board some military vessels. The study is based on the data and information collected by the seven automatic stations, located on the entire Romanian coast and distributed from north to south. Their

location and metadata can be seen in Figure 2 and Table 1, where the altitude represents the station height above the mean sea level, and the sensor altitude represents the sensor height above ground [38].

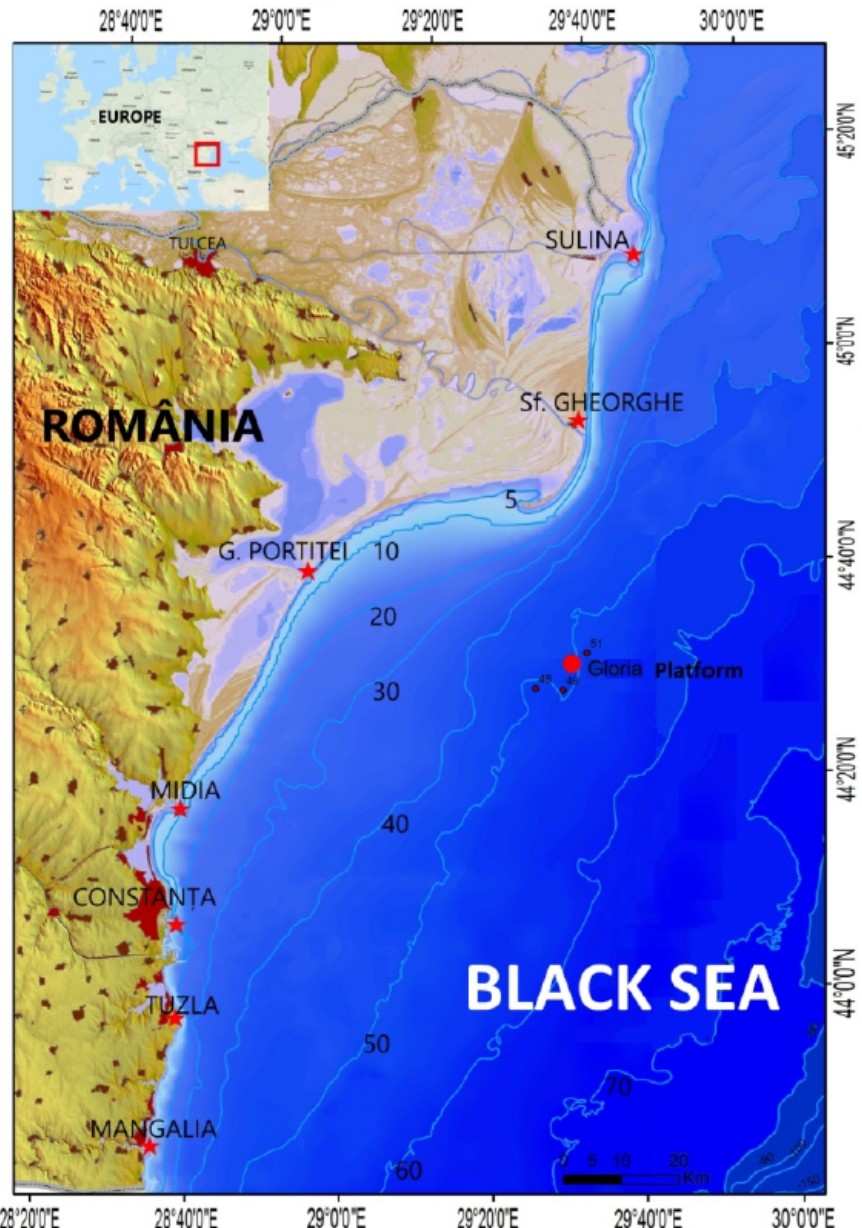

**Figure 2.** Location of weather stations on the Romanian Black Sea coast [39].

Each station is equipped with specific instruments able to process the climate information as detailed in Table 1. Further, the data are transmitted to the MHD headquarters to be registered and analyzed. The system is capable of recording the following data and information: wind speed and direction, air temperature, precipitations, visibility, wind gusts, air pressure, and dew point. The wind parameters which are the key element to our research, are recorded and processed as an average value every 10 min, 24 h a day. The observations were made in the 2009–2021 period.

**Table 1.** Locations and characteristics of the seven automatic weather coastal stations.

| Station | Latitude | Longitude | Altitude | Instruments |
|---|---|---|---|---|
| Sulina | 45.15° | 29.76° | 1.6 m | • Vaisala WXT520 (10 m altitude) heated multisensor composed of:<br>  – WINDCAP ultrasonic wind speed and direction sensor;<br>  – RAINCAP precipitation sensor;<br>  – PTU module including BAROCAP capacitive pressure, THERMOCAP temperature, and HUMICAP humidity sensors.<br>• Vaisala PWD22 (9.7 m altitude) heated present time and visibility multisensor composed of:<br>  – 2 RAINCAP capacitive precipitation sensors;<br>  – optical visibility sensor including PWT11 transmitter and PWC22 receiver/processor;<br>  – temperature sensor—Pt100 thermistor. |
| Sf. Gheorghe | 44.90° | 29.60° | 0.5 m | • Vaisala WXT520 (5.5 m altitude) heated multisensor composed of:<br>  – WINDCAP ultrasonic wind speed and direction sensor;<br>  – RAINCAP precipitation sensor;<br>  – PTU module including BAROCAP capacitive pressure, THERMOCAP temperature, and HUMICAP humidity sensors.<br>• Vaisala PWD22 (5.2 m altitude) heated present time and visibility multisensor composed of:<br>  – 2 RAINCAP capacitive precipitation sensors;<br>  – optical visibility sensor including PWT11 transmitter and PWC22 receiver/processor;<br>  – temperature sensor—Pt100 thermistor. |
| G. Portitei | 44.68° | 28.99° | 0 m | • Vaisala WXT520 (10 m altitude) heated multisensor composed of:<br>  – WINDCAP ultrasonic wind speed and direction sensor;<br>  – RAINCAP precipitation sensor;<br>  – PTU module including BAROCAP capacitive pressure, THERMOCAP temperature, and HUMICAP humidity sensors.<br>• Vaisala PWD22 (9 m altitude) heated present time and visibility multisensor composed of:<br>  – 2 RAINCAP capacitive precipitation sensors;<br>  – optical visibility sensor including PWT11 transmitter and PWC22 receiver/processor;<br>  – temperature sensor—Pt100 thermistor. |
| Midia | 44.32° | 28.69° | 6.3 m | • Vaisala WXT510 (12.7 m altitude) heated multisensor composed of:<br>  – WINDCAP ultrasonic wind speed and direction sensor;<br>  – RAINCAP precipitation sensor<br>  – PTU module including BAROCAP capacitive pressure, THERMOCAP temperature, and HUMICAP humidity sensors. |
| Constanta | 44.15° | 28.67° | 5 m | • Vaisala WS425 (11.2 m altitude) heated ultrasonic wind speed and direction sensor.<br>• Temperature and relative air humidity sensor QMH102, based on the Vaisala HMP45D transmitter (8.2 m altitude), composed of:<br>  – temperature sensor—thermistor Pt-100 IEC 751;<br>  – HUMICAP relative humidity sensor.<br>• Vaisala PWD22 (8.2 m altitude) heated multisensor of present time and visibility composed of:<br>  – 2 RAINCAP capacitive precipitation sensors;<br>  – optical visibility sensor including PWT11 transmitter and PWC22 receiver/processor;<br>  – temperature sensor—Pt100 thermistor.<br>• Vaisala PMT16A pressure sensor—silicon capacitive sensor (7.2 m altitude) |
| Tuzla | 43.99° | 28.67° | 16 m | • Vaisala WXT520 (5.5 m altitude) heated multisensor composed of:<br>  – WINDCAP ultrasonic wind speed and direction sensor;<br>  – RAINCAP precipitation sensor;<br>  – TU module including BAROCAP capacitive pressure, THERMOCAP temperature and HUMICAP humidity sensors. |

**Table 1.** *Cont.*

| Station | Latitude | Longitude | Altitude | Instruments |
|---------|----------|-----------|----------|-------------|
| Mangalia | 43.80° | 28.60° | 3.6 m | <ul><li>Vaisala WS425 (14.3 m altitude) heated ultrasonic wind speed and direction sensor.</li><li>QMH102 air temperature and relative humidity sensor, based on the Vaisala HMP45D transmitter (8.3 m altitude), composed of:<ul><li>temperature sensor—thermistor Pt-100 IEC 751;</li><li>HUMICAP relative humidity sensor.</li></ul></li><li>Vaisala PWD22 (8.2 m altitude) heated multisensor of present time and visibility composed of:<ul><li>2 RAINCAP capacitive precipitation sensors;</li><li>optical visibility sensor including PWT11 transmitter and PWC22 receiver/processor;</li><li>temperature sensor—Pt100 thermistor.</li></ul></li><li>Vaisala PMT16A pressure sensor—silicon capacitive sensor (7.4 m altitude).</li></ul> |

## 3. Results

The present research evaluates the wind data provided by MHD on the 2009–2021 period. Among the reasons for the analysis is to identify an average wind speed value based on the data provided from all seven stations throughout the period. The primary data are represented as average values recorded every day of every month for all 13 years. The evaluation of monthly averaged wind speed data is based on the averages collected from all seven automatic stations each month to analyze the wind patterns and characteristics presented in the Black Sea area. The second analysis was carried out by taking the maximum wind speed value recorded by each station each month during the time interval, then calculating the average of all these maximum values. Additionally, wind roses, which show the frequency and direction of the wind, were created to visualize the data.

Figure 3 and Table 2 provide a first evaluation of the wind speed, emphasizing the distribution of the mean values of each station per month. From the seven stations' data review, we may notice that all data are in the same range, except Sf. Gheorghe, whose values are lower than the averages with approximately 1–2 m/s, because its location is approximately 3 km from the shoreline, the lower wind sensor height (5.5 m), and the higher surface roughness due to surrounding terrain whereas all the other stations are situated on the shoreline (Gura Portitei, Tuzla), or even slightly in the sea (Sulina, Midia, Constanta, Mangalia) and have the sensors at about 10 m.

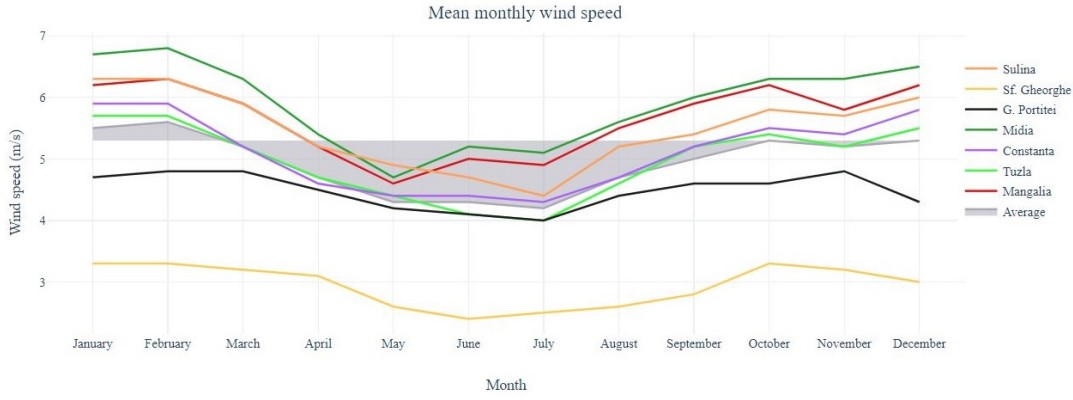

**Figure 3.** Mean monthly values of the wind speed.

**Table 2.** Mean monthly values of the wind speed.

|  | Jan | Feb | Mar | Apr | May | Jun | Jul | Aug | Sep | Oct | Nov | Dec |
|---|---|---|---|---|---|---|---|---|---|---|---|---|
| Constanta | 5.9 | 5.6 | 5.2 | 4.6 | 4.4 | 4.4 | 4.3 | 4.7 | 5.2 | 5.5 | 5.4 | 5.8 |
| Mangalia | 6.2 | 6.2 | 5.9 | 5.2 | 4.6 | 5.0 | 4.9 | 5.5 | 5.9 | 6.2 | 5.8 | 6.2 |
| Sulina | 6.3 | 6.3 | 5.9 | 5.2 | 4.9 | 4.7 | 4.4 | 5.2 | 5.4 | 5.8 | 5.7 | 6.0 |
| G. Portitei | 4.7 | 5.0 | 4.8 | 4.5 | 4.2 | 4.1 | 4.0 | 4.4 | 4.6 | 4.6 | 4.8 | 4.3 |
| Sf. Gheorghe | 3.3 | 3.6 | 3.2 | 3.1 | 2.6 | 2.4 | 2.5 | 2.6 | 2.8 | 3.3 | 3.2 | 3.0 |
| Midia | 6.7 | 6.6 | 6.3 | 5.4 | 4.7 | 5.2 | 5.1 | 5.6 | 6.0 | 6.3 | 6.3 | 6.5 |
| Tuzla | 5.7 | 5.6 | 5.2 | 4.7 | 4.4 | 4.1 | 4.0 | 4.6 | 5.2 | 5.4 | 5.2 | 5.5 |
| Average | 5.5 | 5.6 | 5.2 | 4.7 | 4.3 | 4.3 | 4.2 | 4.7 | 5.0 | 5.3 | 5.2 | 5.3 |

As can be seen, the most energetic month is February, with an average wind speed of 5.6 m/s, followed by January (5.5 m/s), and both December and October, with a value of 5.2 m/s. The outcome of the presented analysis reveals that the most dynamic season is winter. The following season with great value is autumn. As presented earlier, October reported a mean wind speed of 5.3 m/s, close to November (5.2 m/s) and September (5 m/s). Further, the next season evaluated is spring, with the following wind speed recorded: March—5.2 m/s, April—4.7 m/s, and May—4.3 m/s. Lastly, the lowest energetic season is summer; the lowest value registered belongs to July—4.2 m/s, followed by June—4.3 m/s, and August—4.7 m/s.

From the analysis made, based on the entire data collected over the 13-year period, an average wind speed value of 4.9 m/s was obtained.

The highest average values of wind speed were recorded by Midia station (5.9 m/s), situated on the south part of the Romanian coast, followed by Mangalia (5.6 m/s), also located in the south, Sulina—5.5 m/s (north), Constanta—5.1 m/s (south), Tuzla—5 m/s (south), G. Portitei—4.5 m/s (north), and Sf. Gheorghe—3 m/s (north), resulting in greater values registered in the southern part of the Romanian coast.

In Figure 4, a more precise analysis of the average and maximum wind speed of each month can be observed, based on the information collected from all seven automatic stations during the entire period.

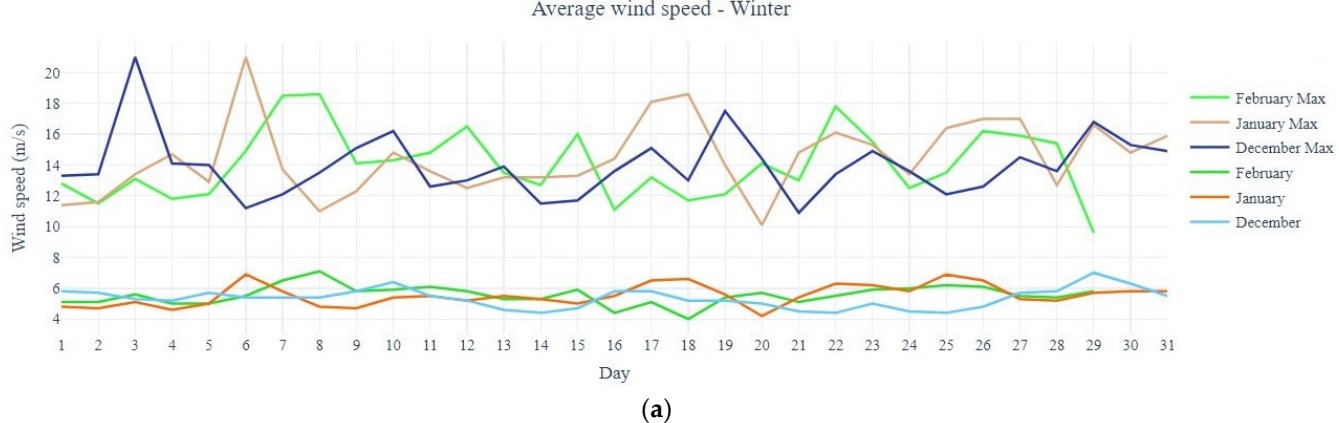

(**a**)

**Figure 4.** *Cont.*

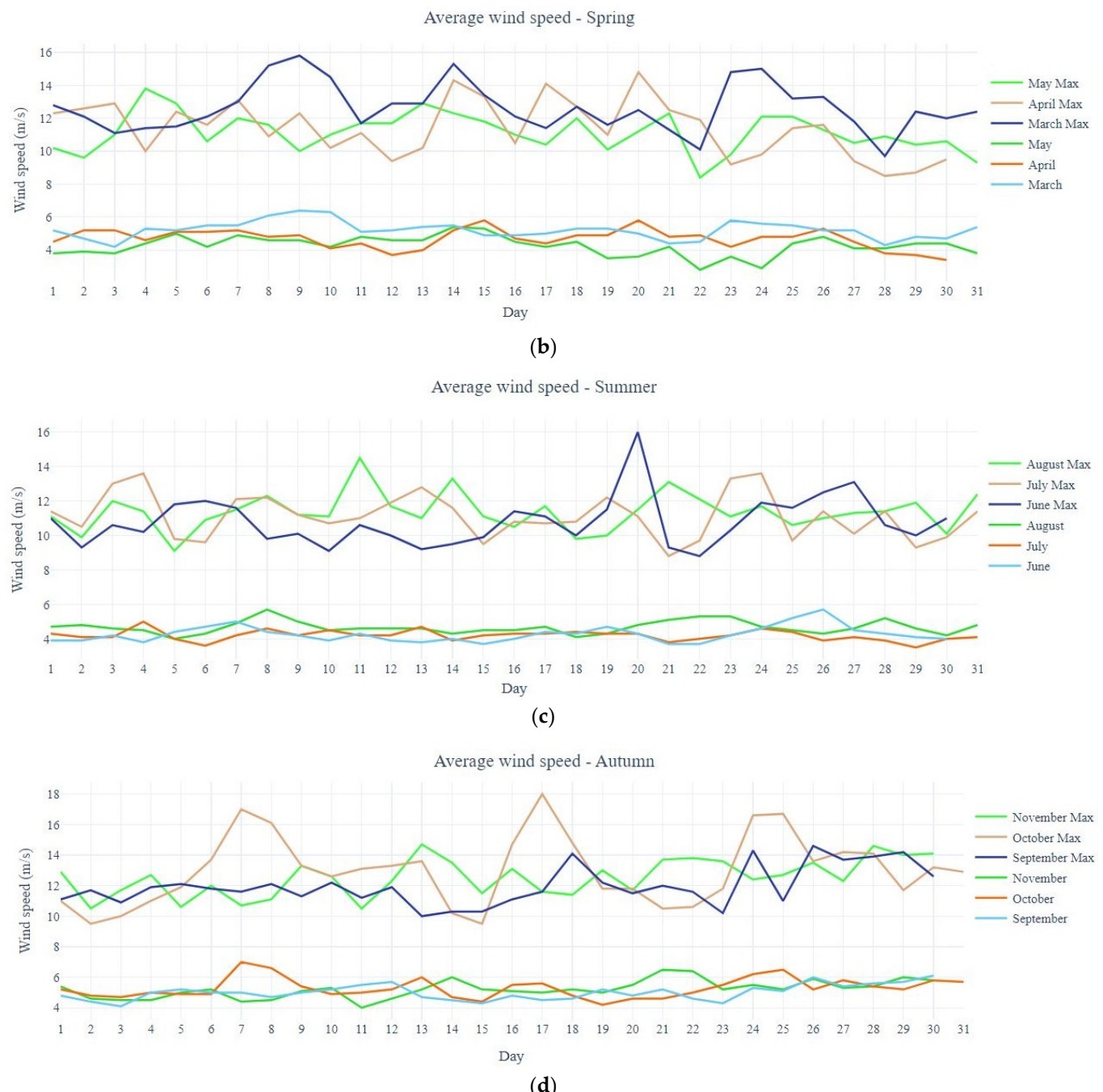

**Figure 4.** Monthly wind speed analysis; (**a**) winter; (**b**) spring; (**c**) summer; (**d**) autumn.

### 3.1. Mean Wind Data Analysis per Month

In the present section, the mean wind speeds each month were analyzed to outline in a comprehensive way the climatic picture of the Romanian coast of the Black Sea. As presented above, January is one of the most energetic months of the year, alongside February and December, with a registered wind speed mean value of 5.5 m/s. The values recorded in this month are in good agreement, being in the range of 4.2 m/s (20 Jan) and 6.9 m/s (two observations made on 6 and 25 Jan). Following the same trend, in February, mean wind speeds were recorded in the range of 4 m/s (18 Feb) and 7.1 m/s (8 Feb), having an average wind speed of 5.6 m/s. The maximum average wind speed from February represents the greatest value in this particular analysis. From the evaluation made, February is the most dynamic month; nevertheless, no major variations were observed. The next month

presented is March, where an average of 5.2 m/s was recorded. The maximum value was recorded on the 9th—6.4 m/s, whereas the lowest was on the 3rd—4.2 m/s. In the fourth month of the year, values in the range of 3.4 (30 Apr)—5.8 m/s (15 and 20 Apr) with a mean value of 4.7 m/s were registered. Spring season evaluation ends in May, which has an average wind speed data of 4.3 m/s. From the analysis made it can be seen that a maximum value of 5.4 m/s was noticed on the 14th, whereas a minimum of 2.8 m/s was on the 22nd, representing the lowest value recorded regarding the average wind speed. The evaluation of the data collected from the seven stations illustrates a continuous decrease with the summer season, where the lowest values were recorded. Based on the evaluation made, in June a mean wind speed of 4.3 m/s was noticed. In the present month, minimum values of 3.7 m/s (three observations made on 15, 21, and 22) and a maximum of 5.7 m/s on the 26th were registered. July is the lowest energetic month with a wind speed value of 4.2 m/s. Several minimum values were recorded: 3.5 m/s—30th, 3.6 m/s—6th, and also a maximum of 5 m/s on 4th. In terms of wind variations, July is a stable month with differences between the maximum and minimum of 1.4 m/s. In addition, August wind values have differences of 1.7 m/s between the maximum registered (5.7 m/s—8 Aug) and the minimum (4 m/s—6 Aug). The next season discussed is autumn, with the following months: September, October, and November. Firstly, September shows an average of 5 m/s, a minimum of 4.1 m/s at the beginning of the month (3 Sep), and a maximum of 6.1 m/s at the end of the month (30 Sep). Secondly, in October, values in the range of 4.2 m/s (19 Oct) and 7 m/s (7 Oct) were recorded. The average wind speed value noticed was 5.3 m/s, close to the one observed in November—5.2 m/s. It can be noticed that values are increasing as the months approach the end of the year. A minimum observation of 4 m/s was made on 11th November, whereas a maximum observation was made on the 21st—6.5 m/s. The last month of the present analysis, December, shows an average of 5.3 m/s, outlining once more that it is a dynamic month concerning the wind energy potential. The analysis outlines a maximum of 7 m/s on the 29th and minimum values of 4.4 m/s on the 14th, 22nd, and 25th. Based on the evaluation made for each month during the entire period analyzed, it resulted that the values were encountered in the range of 2.8 m/s (22nd of May) and 7.1 m/s (8th of February). Nevertheless, no major variations were encountered, and the values obtained followed either an increasing or decreasing path.

*3.2. Maximum Wind Values*

The second approach of the present paper represents the analysis of the maximum values obtained from the evaluation of wind speed. The maximum wind values in the Black Sea can vary significantly depending on the season and the specific location. In general, the Black Sea is known for its strong winds, especially during the winter months when storms are more common. The highest wind speeds are typically found in the western part of the Black Sea, whereas in the eastern part of the Black Sea, the winds are generally less strong. It is important to note that the Black Sea is also prone to sudden and severe storms, which can bring strong winds and rough seas. These storms can occur at any time of year, but they are most common in the winter and spring.

In Figure 5 and Table 3, the monthly maximum wind speed broken down by stations is illustrated. The outline revealed from the entire database suggests a mean maximum value of 12.4 m/s. The highest mean value registered was in January—14.5 m/s, followed by February—14.1 m/s and December—14 m/s, resulting again that winter is the most intense season concerning the wind potential. The next season with dynamic conditions is autumn, with the following values: October—13 m/s, November—12.5 m/s, and September—11.9 m/s. Following the same criteria, the third season is spring. In March, a maximum mean wind speed of 12.6 m/s was registered, whereas in April it was 11.4 m/s, and in May 11.1 m/s. The last season discussed is summer with the following recorded values: August—11.4 m/s, July—11.1 m/s, and the lowest high value—10.8 m/s (June). This analysis follows the same trend as the mean wind speed with the same season's order depending on the wind dynamic.

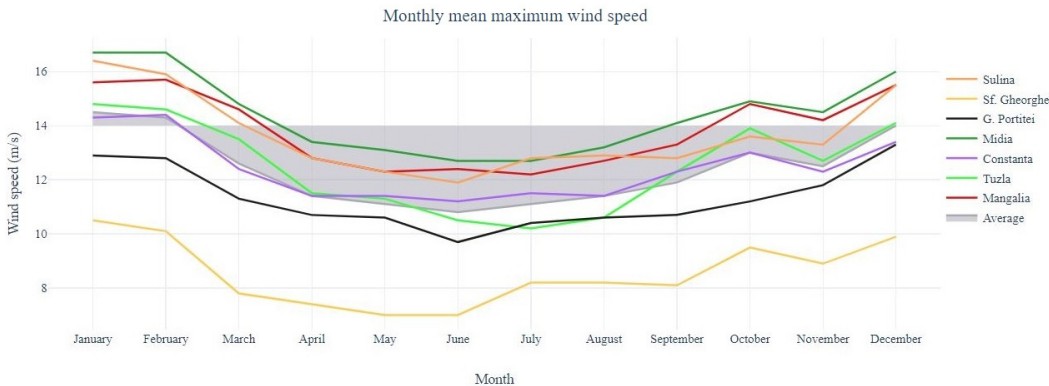

**Figure 5.** Monthly mean maximum values of the wind speed.

**Table 3.** Monthly mean maximum values of the wind speed.

|  | Jan | Feb | Mar | Apr | May | Jun | Jul | Aug | Sep | Oct | Nov | Dec |
|---|---|---|---|---|---|---|---|---|---|---|---|---|
| Constanta | 14.3 | 13.8 | 12.4 | 11.4 | 11.4 | 11.2 | 11.5 | 11.4 | 12.3 | 13.0 | 12.3 | 13.4 |
| Mangalia | 15.6 | 15.7 | 14.6 | 12.8 | 12.3 | 12.4 | 12.2 | 12.7 | 13.3 | 14.8 | 14.2 | 15.5 |
| Sulina | 16.4 | 15.8 | 14.1 | 12.8 | 12.3 | 11.9 | 12.8 | 12.9 | 12.8 | 13.6 | 13.3 | 15.5 |
| G. Portitei | 12.9 | 12.5 | 11.3 | 10.7 | 10.6 | 9.7 | 10.4 | 10.6 | 10.7 | 11.2 | 11.8 | 13.3 |
| Sf. Gheorghe | 10.5 | 10.1 | 7.8 | 7.4 | 7.0 | 7.0 | 8.2 | 8.2 | 8.1 | 9.5 | 8.9 | 9.9 |
| Midia | 16.7 | 16.3 | 14.8 | 13.4 | 13.1 | 12.7 | 12.7 | 13.2 | 14.1 | 14.9 | 14.5 | 16.0 |
| Tuzla | 14.8 | 14.4 | 13.5 | 11.5 | 11.3 | 10.5 | 10.2 | 10.6 | 12.3 | 13.9 | 12.7 | 14.1 |
| Average | 14.5 | 14.1 | 12.6 | 11.4 | 11.1 | 10.8 | 11.1 | 11.4 | 11.9 | 13.0 | 12.5 | 14.0 |

Moreover, it is concluded that the maximum wind speed values recorded from each station are roughly in the same sequence as presented earlier in the highest average values evaluation, reiterating that the southern part presents high values than the northern part: Midia—14.4 m/s, Mangalia—13.8 m/s, Sulina—13.7 m/s, Tuzla—12.5 m/s, Constanta—12.4 m/s, G. Portitei—11.3 m/s, and Sf. Gheorghe—8.6 m/s.

*3.3. Maximum Wind Values Recorded per Month*

In this section, the averages of the greatest wind speeds recorded per month are analyzed to illustrate the most dynamic periods referred to in this study. As explained earlier, with a value of 14.5 m/s, January is the month with the highest data registered in terms of mean maximum wind speeds. The peak of this analysis is reached on the 6th when a value of 21 m/s was registered. The same value is encountered on the 3rd of December. Considerable great values were seen in January: 18 Jan—18.6 m/s, 17 Jan—18.1 m/s, and 26 and 27 Jan—17 m/s. Close to the monthly maximum mean speed recorded in January is the one registered in February—14.1 m/s. Apart from the peak recorded on the 8th of February (18.6 m/s), other high values were recorded on the dates of the 7th (18.5 m/s) and 22nd (17.8 m/s), but after these values, the lowest data was observed—9.6 m/s (29 Feb). Further, moving to the next season, high values are considerably lower. In March a mean maximum value of 15.8 m/s was marked on the 9th, whereas the average value is 12.6 m/s. Other great values were seen on the following dates: 15.3 m/s—14 Mar, 15.2 m/s—8 Mar, and 15 m/s—24 Mar. Particular attention is paid to April, which presents an extended period with greater values than the rest of the month. In the period of 13–22 April appropriate wind data were observed resulting in stable conditions in terms of weather parameters. Peaks of 14.3 m/s on the 14th, 14.1 m/s on the 17th, and 14.8 m/s on the 20th were recorded, whereas medium values were seen among them in order to smooth the continuous wind trend. In May no major variations were observed. Values between 8.4 m/s (22nd of May)

and 13.8 m/s (4th of May) were seen, as in Figure 4b. In June, the lowest mean value concerning the maximum wind speed was observed—10.8 m/s. In this month, a peak of 16 m/s was seen on the 20th, and others in the range of 8.8 m/s (22 Jun) and 13.1 m/s (27 Jun) were recorded. Low values are also detailed in Figure 4c. In July, maximum wind average speeds are considerably lower: 13.6 m/s—4 Jul and 24 Jul. The last month of the summer season, August presents an average maximum wind speed of 11.4 m/s. A maximum value of 14.5 m/s was recorded on the 11th; still, no major variations were observed. The second energetic season is autumn. In September all values are in good agreement, being in the range of 10 m/s (13th Sep) and 14.6 m/s (26th Sep). In October, values are considerably higher, recording a mean maximum wind speed of 13 m/s. Other high values presented in this month can be seen in Figure 4d (18 m/s—17 Oct, 17 m/s—7 Oct, 16.7 m/s—25 Oct). November represents a stable month, illustrating a difference of 4.2 m/s between the lowest value (10.5 m/s—2 Nov and 11 Nov) and 14.7 m/s—13 Nov. Similar wind data are seen in this month with minor ups and downs. The last month analyzed is December. Like February, December presents significant variations in terms of mean maximum wind speeds with differences of 10.1 m/s observed. As presented earlier, a peak of 21 m/s was seen on the 3rd of December, followed by lower values: 17.5 m/s—19 Dec and 16.8 m/s—29 Dec. The analysis made in this section concluded that winter is the most suitable season in terms of dynamic wind parameters, nevertheless unstable conditions are seen in December and February. Stable conditions are located in November and April.

*3.4. Wind Directions*

The wind direction in the Black Sea can vary greatly depending on the time of year and the specific location within the sea. The Black Sea is generally influenced by the prevailing winds in the region, typically westerly or southwesterly. During the winter months, the winds tend to be stronger and more consistent, whereas, in the summer, the winds are generally lighter and more variable. In addition to the prevailing winds, the Black Sea is also influenced by local weather patterns and topographic features, which can affect wind direction and strength. The purpose of the present section is to evaluate the wind frequencies by direction, depending on the month (Figure 6).

The wind direction is typically from the northwest in the winter and from the southeast in the summer, however, a few exceptions have been found based on the data used on the time interval. The winter months are basically following the same wind frequencies: December—NW (22.7%) and W (22.6%); January—NW (21.8%) and N (19.7%); February—N (22.9%), S—(15.5%). In spring, great wind frequencies are found from the southeast. In March, there are high values both in the north (19.3%) and south (17.4%), representing a transit month from one season to another. In April and May, great values are found from the south (19.4%—Apr, 17%—May), followed by wind frequencies from the southeast (18.3%—Apr, 15.8%—May). In June, winds from the south can be found with a frequency of 16.4%, followed by those from the north—15.8%. In July, the wind shifts to NW (18%) and N (16.1%). August, the last month of summer, has similar values related to July: N (23.6%) and NW (18%). The wind directions presented in autumn are from the northwest: September—N (18.5%), NW (15.9%), October—N (21.1%), NW (16.9%), and November—N (20.6%), NW (16.7%). The analysis highlighted that during winter and autumn seasons the prevailing wind direction is from N–NW–W, whereas in spring the wind blows more frequently from S–SE or N. The summer season presents higher wind direction frequencies from N–NW or S.

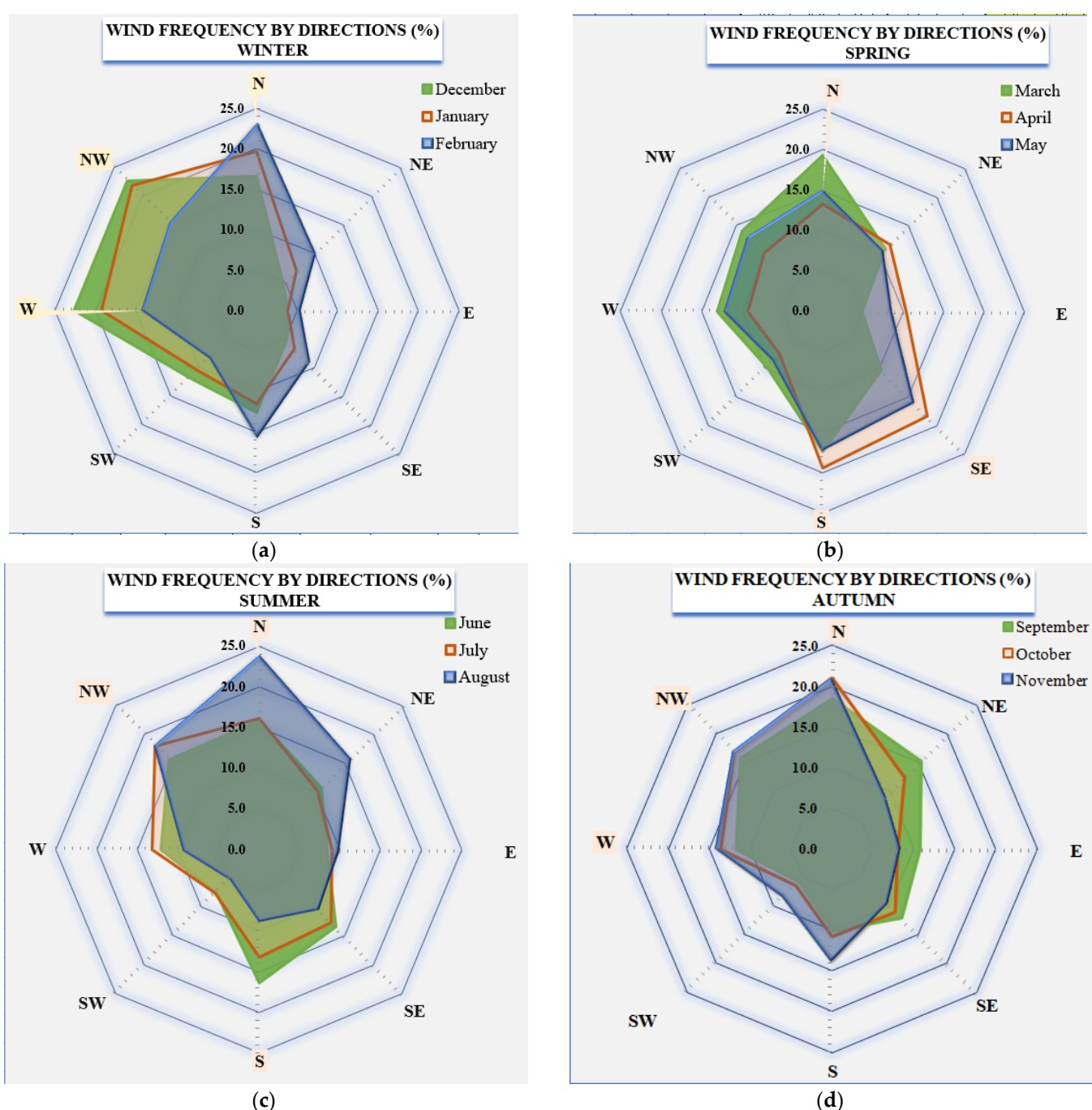

**Figure 6.** Wind frequency by directions; (**a**) winter; (**b**) spring; (**c**) summer; (**d**) autumn.

## 4. Discussion

This study aimed to assess the climatic wind picture along the western coast of the Black Sea, identifying its mean and maximum wind speed values each month to provide the most suitable periods in terms of wind energy potential. Moreover, the wind directions were analyzed. The outcome of this research is based on the MHD weather data over a 13-year interval (2009–2021) from seven automatic marine stations situated on lighthouses along the Romanian Black Sea coast.

An important aspect that requires more detailed investigation is the storm dynamics presented in the Black Sea. Storm dynamics in the Black Sea are complex phenomena and can be influenced by various factors. One important factor is the contrast in temperature between the sea and the surrounding land, creating instability in the atmosphere, which

can lead to the development of storms. Another crucial factor is the movement of air masses. Warm, moist air from the Mediterranean Sea can move northward into the Black Sea, interacting with colder air masses from the east or west, leading to the development of storms, particularly in winter and spring.

The aim of this analysis is to identify the periods that presented storm characteristics (wind speed greater than 14 m/s). The research was based on the average stormy days from all seven stations over the 13-year interval. In January, an average of 28 stormy days was registered, the most significant dates being 3, 6, 17, 18, 22, 25, 26, 27, and 30 January. Midia station reported 52 days in January with great wind values in the evaluated period. In February, a lower value was registered—25 stormy days and once again Midia station recorded the most numerous days (45). The following days were observed with high values: 3, 17, 18, 22, 25, and 26 February. Further, in spring, the number of days with storm aspects decreases: March—12 days, April—8 days (the 20th of April represents the most energetic day concerning the storms occurring), and May—4 days. No extremes were seen in summer. In all three months of this season (June, July, and August) a value of 3 days per month was seen in terms of the average number of stormy days. Moving to the autumn season, the number of stormy days is significantly increasing. In September an average of 8 days with wind speed greater than 14 m/s was registered, Midia being again the station with the most numerous days registered (23). The average value obtained in October was 17 days which fulfilled the storm criteria, and various days were seen with high values (7, 8, 16, 17, 24, and 25 of October). November registered an average number of stormy days of 11 days. Lastly, based on the data from all seven stations, an average of 23 stormy days was obtained in December. Midia registered 45 days, whereas Mangalia obtained 43 days for the total period evaluated. The following days are noticed as dynamic ones (10, 29, 30, and 31 of December). Based on the evaluation, it is concluded that the winter months present the tendency for storms to occur (December—23 days, January—28 days, February—25 days). Moreover, it is revealed that these complex weather conditions are fulfilled in the southern part of the analyzed area. Still, no significant variations are noticed.

## 5. Conclusions

In the present research, the wind potential of the Romanian coast of the Black Sea was evaluated considering its main parameters (speed and direction). Based on the MHD wind datasets considering a 13-year time interval, the wind climate of the respective area was highlighted, indicating the most suitable months for strong and persistent winds. Relevant trends considering the wind dynamics were outlined, resulting from the evaluation of mean and maximum wind speed values. In addition, the wind directions issue was discussed and a summary regarding the periods in which present storm characteristics was elaborated.

From the analysis made, based on the entire data collected over the 13-year period, an average wind speed value of 4.9 m/s and a mean maximum value of 12.4 m/s were obtained. All the analyses made indicate that during winter, which is the most energetic season, strong winds are encountered and the most numerous days with storms are seen; nevertheless, unstable conditions are noticed in December and February. Moreover, it is revealed that wind speed values recorded from each station are roughly in the same sequence, although greater values were registered in the southern part of the Romanian coast. Concerning the wind directions, it was highlighted that most wind directions are from N or NW regardless of the season (winter, summer, or autumn), whereas other directions (S or SE) were seen in summer.

The Black Sea is known for its variable and sometimes unpredictable weather, including strong winds and storms, particularly during winter. These winds can reach high speeds and can create large waves and rough sea conditions. The wind's direction and strength can affect the sea's navigability and shipping safety. Strong winds and rough seas can make it difficult for vessels to maintain their course and can cause delays in shipping. On the other hand, favorable wind conditions can help to speed up the voyage and make it more efficient [40]. In conclusion, the offshore wind industry presented in

Europe has been rapidly increasing in recent years, with many nations eager to invest heavily in the development of offshore wind farms. However, this area is not considered an ideal location for offshore wind power due to its generally lower wind speeds than those in other European areas where offshore wind power is more prevalent, such as the North Sea and the Baltic Sea. Although, the present research proves that the western coast of the Black Sea has moderate to high wind resources and may be suitable for nearshore wind power generation. To sum up, more research is needed to fully understand the wind characteristics of the Black Sea and the potential for offshore or nearshore wind power in this region.

**Author Contributions:** Writing—review and editing, methodology, L.-I.N.; data curation, resources V.-M.T.; supervision and project administration, E.R. All authors have read and agreed to the published version of the manuscript.

**Funding:** This work was carried out in the framework of the research project CLIMEWAR (CLimate change IMpact Evaluation on future WAve conditions at Regional scale for the Black and Mediterranean seas marine system), supported by a grant of the Ministry of Research, Innovation, and Digitization, CNCS—UEFISCDI, project number PN-III-P4-PCE-2021-0015, within PNCDI III.

**Institutional Review Board Statement:** Not applicable.

**Informed Consent Statement:** Not applicable.

**Data Availability Statement:** The data that were used in the present study are available by official statement. The wind data were obtained from the Maritime Hydrographic Directorate.

**Conflicts of Interest:** The authors declare no conflict of interest.

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
