# Peer review of "An Evaluation of the Wind Energy along the Romanian Black Sea Coast"

_inventions, doi:10.3390/inventions8010048_

Round 1

Reviewer 1 Report

Review of “An evaluation of the wind energy along the Romanian coast of the Black Sea” by Nedelcu and Rusu

[self-plagiarism concerns]

I feel annoyed when I see the significant similarities between this manuscript and “An Analysis of the Wind Parameters in the Western Side of the Black Sea” by the same authors, published in the same journal about a year ago. https:// doi.org/10.3390/inventions7010021. 

The authors need to specify how the current manuscript stands out from their previous publications. Otherwise, it’s self-plagiarism misconduct. 

[suitability of the journal]

Also, I am not sure how the topic can fit the inventions journal. What is being invented?

The figure qualities can be improved as well.

[quality in data presentation]

For Figs. 3 and 4, please consider using a subplot and keeping the original aspect ratios.

For Figs. 5, for wind-rose graphs, you can group multiple months in a plot with different line-style/line-color combinations with a fixed scale. Do not use Excel to plot the figures, or change the color schemes to a publishing-friendly set.

[data access]

Please release all original raw data.

Reviewer 2 Report

The authors evaluated the wind energy along the Romanian coast of the Black Sea using observed wind data at seven wind stations over a 13-year time period. Averaged wind velocity and direction per month show monthly variation of wind energy. The results provide good reference for sea’s navigability and shipping’s safety in the Black Sea. I have some specific questions for the authors’ answer as follows:

1.     Is the altitude at Tuzla 16m in Table1?

2.     Line 173-188; Activities of the Maritime Hydrographic Directorate are rarely related to the wind dataset. It is recommended to delete this part.

3.     Line 213-218; Wind motion is complex and varies with environmental factors. Topographical roughness makes wind to have vertical distribution. The anemometer of seven stations are located at different altitude. Fig.3 shows the mean wind speed of each station per month. The lowest values are found at Sf. Gheorghe. The authors attribute the location of the station far from onshore. However, comparing the two stations with the highest average value and the two stations with the lowest value, it is obvious that the wind speed in this area is affected by both the measurement height and the local topography. The reason for the lowest wind speed at the Sf. Gheorghe station needs to be clarified.

4.     What is the statistical or engineering application of the daily averaged and or maximum wind speed in Figure 4?

5.      In the introduction, the current planning and development of offshore wind power in Europe and the wind characteristics of the Black Sea are described. However, the conclusion shows that the research results of this paper can be applied to shipping’s safety. The research results must meet the research objectives.
